# Chemical Characterization of *Trichoderma* spp. Extracts with Antifungal Activity against Cocoa Pathogens

**DOI:** 10.3390/molecules28073208

**Published:** 2023-04-04

**Authors:** Ivan Chóez-Guaranda, Fernando Espinoza-Lozano, Dennys Reyes-Araujo, Christian Romero, Patricia Manzano, Luis Galarza, Daynet Sosa

**Affiliations:** 1Centro de Investigaciones Biotecnológicas del Ecuador, ESPOL Polytechnic University, Escuela Superior Politécnica del Litoral, ESPOL, Guayaquil P.O. Box 091050, Ecuador; 2Departamento de Ciencias de la Vida y de la Agricultura, Universidad de las Fuerzas Armadas-ESPE, Sangolquí P.O. Box 171-5-231B, Ecuador; 3Facultad de Ciencias de la Vida, ESPOL Polytechnic University, Escuela Superior Politécnica del Litoral, ESPOL, Guayaquil P.O. Box 091050, Ecuador

**Keywords:** biocontrol, frosty pod rot, moniliasis, *Moniliophthora*, witches’ broom

## Abstract

Ecuador is one of the major cocoa producers worldwide, but its productivity has lately been affected by diseases. Endophytic biocontrol agents have been used to minimize pathogenic effects; however, compounds produced by endophytes are minimally understood. This work presents the chemical characterization of the *Trichoderma* species extracts that proved inhibition against cocoa pathogens. Solid–liquid extraction was performed as a partitioning method using medium with the fungal mycelia of *Trichoderma reesei* (C2A), *Trichoderma* sp. (C3A), *Trichoderma harzianum* (C4A), and *Trichoderma spirale* (C10) in ethyl acetate individually. The extract of *T. spirale* (C10) exhibited the growth inhibition (32.97–47.02%) of *Moniliophthora perniciosa* at 10 µg/mL, while a slight stimulation of *Moniliophthora roreri* was shown by the extracts of *T. reesei* (C2A) and *T. harzianum* (C4A) at higher concentrations. The inhibitory activity could be related to alkaloids, lactones, quinones, flavonoids, triterpenes, and sterols, as indicated by chemical screening and antifungal compounds, such as widdrol, *β*-caryophyllene, tyrosol, butyl isobutyrate, sorbic acid, palmitic acid, palmitelaidic acid, linoleic acid, and oleic acid, which were identified by gas chromatography–mass spectrometry (GC-MS). The results showed that the extracts, particularly *T. spirale* (C10), have the potential as biocontrol agents against witches’ broom disease; however, further studies are needed to confirm their effectiveness.

## 1. Introduction

Aging plantations, degraded soils, pests, and diseases have been described as the main causes that affect cocoa (*Theobroma cacao*) productivity worldwide, directly influencing cocoa farmers and the cocoa industry in the tropical regions of South East Asia and West Africa, as well as Central and South America [1]. In Ecuador, cocoa production is an important trade that supports the country’s economy and provides employment for thousands of people. Ecuador is the third-largest producer of cocoa in the world after the Ivory Coast and Ghana, and the biggest producer in Latin America, with an estimated production of over 365,000 metric tons in 2021 [2]. Nevertheless, the fungal diseases frosty pod rot (*Moniliophthora roreri*) and witches’ broom (*Moniliophthora perniciosa*, formerly *Crinipellis perniciosa*) have adversely affected cocoa plantations, causing significant economic losses due to the decrease in production yields [3]. The fungus *M. roreri* can grow in different environmental conditions causing different symptoms in a complete cycle of about 183 days. Sometimes, cocoa pods are seemingly healthy and continue normal growth but are internally infected and develop disease symptoms later in development. The typical symptoms are necrotic flecks on the surface of cacao pods, which can be observed in the 60–80 days old pods. Thereafter, a dark brown stain with an irregular border that entirely covers the pod, pod malformation, and internal damage characterized by a dark brown fine powder (spores) over the cocoa beans and pulp can be observed when the infection has been developing for more than 110 days [4]. *M. perniciosa* has multiple penetration modes, mainly through trichomes and stomata. It can cause infection in different plant organs, such as flowers, pods, and branches. The typical symptoms include abnormal shoots that display stem swellings, bud proliferation, and vegetative brooms, which inspired the name witches’ broom disease in cocoa [5]. The results of recent studies from the major cocoa-producing provinces of Ecuador showed that *Moniliophthora* species possess high morphological and genetic diversity [6,7,8,9]. For this reason, some alternatives such as cacao agroforestry systems [1] and biological controls using products based on endophytic organisms, including the combined application of natural or chemical products [10], have been used to reduce the impact of diseases, minimize the utilization of fungicides, and improve the yield of cocoa production.

In this context, *Trichoderma* comprises a diverse fungi group of more than 100 species, which have been identified using molecular techniques [11]. It is the most studied and marketed endophyte fungi, since it has been used as a potent biocontrol agent and plant growth promoter [12,13]. Its remarkable biological activity has demonstrated effectiveness against other fungi and plant pathogens, including bacteria, viruses, and nematodes, using antagonistic actions based on antibiosis, competition, and mycoparasitism [14,15]. Furthermore, the species of the *Trichoderma* genus can induce systemic resistance (ISR) in plants. ISR is a form of immunization in which host plants become resistant to pathogens. This resistance is induced by the application of natural compounds produced by *Trichoderma*, which activate the plant’s immune system, causing the production of defense proteins and other molecules that can fight off invading pathogens [16]. In fact, many secondary metabolites have been reported for different culture conditions, including volatile and non-volatile compounds during the in vitro interaction of different *Trichoderma* species with plants and other microorganisms [17,18,19] and the improvement in the production of target compounds using genetic engineering to increase metabolite production [20]. In particular, several studies from Mexico [21], Panama [22], Peru [23,24], and Brazil [25] have evidenced *Trichoderma–Moniliophthora* interactions. In Ecuador, the antagonism of *Trichoderma* species from different regions has shown inhibition against banana and cocoa pathogens [26,27]. Nevertheless, the responsible compounds of the biological activity have been rarely described. 

Therefore, this study addresses the chemical characterization of four Ecuadorian *Trichoderma* spp. Extracts, which have been proven to inhibit the growth of the cocoa pathogens *M. roreri* and *M. perniciosa*, in order to determine potential specific applications that benefit cocoa cultivars.

## 2. Results and Discussion

### 2.1. Antifungal Activity

The effect of ethyl acetate extracts on the assessed cocoa pathogens, *M. perniciosa* and *M. roreri,* is presented in Figure 1 and Figure 2, respectively. 

The antifungal activity of metabolites produced by *Trichoderma* strains was expressed as the percentage of growth inhibition (PGI). The results showed that *M. perniciosa* was the most sensitive pathogen species (Figure 3). The maximum inhibition was observed for the extract of *T. spirale* (C10) at 10 µg/mL (47.02 ± 1.17%), followed by *T. harzianum* (C4A) at 10 µg/mL (39.04 ± 2.43%). In addition, the extract of *T.* sp. (C3A) presented the highest inhibition (36.07 ± 0.55%) at the minimum concentration of 1 µg/mL; however, no significant differences were detected among evaluated *Trichoderma* strains at this concentration. Even though the precise data on the *T. spirale–M. perniciosa* interaction are limited, higher inhibitory activity (75%) affecting mycelial growth and even spore germination of *M. perniciosa* has been reported for *T. harzianum* [28], and significant mycoparasitism levels against *M. roreri* have been achieved by *T. spirale* [29]. In addition, remarkably, antifungal activity against *M. perniciosa* and *M. roreri* has been found for foliar endophytic fungi [30] and endophytic *Bacillus* [31] isolated from cocoa cultivars. Additionally, *T. spirale* has been shown to inhibit the growth of *Fusarium oxysporum* during in vitro and greenhouse experiments, demonstrating the relevance of the metabolites produced by this strain, which also indicates plant protection against fungal disease [32].

On the other hand, the best PGI against *M. roreri* was observed for the extract of *T. spirale* (C10) at 1000 µg/mL (26.20 ± 2.06%) and 100 µg/mL (17.10 ± 1.23%). Higher PGI values (48%) have been reported for culture filtrates from Brazil [31] and comparable data (15.90–37.10%) for isolates from Mexico [20] against *M. roreri*. Differences among results could be attributed to the technique employed since the referenced studies used a dual-culture assay, through which the inoculum size, agar volume, well size, and other variables possibly affect the fungal interactions [17], isolates, culture filtrates, extracts, or compounds assessed. Contrasting results were shown for the extracts of *T. reesei* (C2A) (4.40%) and *T. harzianum* (C4A) (4.36%) at 1000 µg/mL. Additionally, the extracts of *T.* sp. (C3A) (0.49%) and *T. spirale* (C10) (1.45%) stimulated the growth of the pathogen at 100 and 1 µg/mL, respectively. The growth stimulation could be explained by the metabolites of the extracts, which, at higher concentrations, seem to favor the pathogen. Similar results have been reported for endophytic fungal isolates, demonstrating the growth promotion of *Moniliphthora* spp. Fungal endophytes produce several compounds, such as plant growth hormones (auxins, cytokinins, and gibberellins), which can also promote the growth of fungi [33]. Different findings have evidenced the growth restraint of *M. roreri* through *T. harzianum* (40.50–41.20%), *T. ressei* (20.30–25.80%) in Mexico [20], *T. harzianum* (75%) in Ecuador [27], and native *Trichoderma* strains (48.50–57.94%) in Peru [23]. Even so, *M. roreri* growth inhibition (25–97%) and growth stimulation (50–140%) ranges have been indicated by the fungicide flutolanil [32].

These results show the importance of identifying the metabolites secreted by *Trichoderma* species, since the extracts revealed similar properties as in the aforementioned commercial antifungal agent.

### 2.2. Chemical Screening 

The content of dried ethyl acetate extracts was *T. reesei* (C2A) (12.60 g/L), *T.* sp. (C3A) (5.50 g/L), *T. harzianum* (C4A) (8.20 g/L), and *T. spirale* (C10) (5.30 g/L). In general, the chemical screening of *Trichoderma* spp. ethyl acetate extracts revealed the presence of alkaloids, lactones, quinones, reducing sugars, flavonoids, triterpenes, and sterols (Table 1). It should be noted that the extract of *T. reesei* (C2A) showed positive results for most of the secondary metabolites tests, followed by *T. spirale* (C10), *T. harzianum* (C4A), and *T.* sp. (C3A). Alkaloids were detected only in the extracts of *T. reesei* (C2A) and *T. harzianum* (C4A). The presence of flavonoids was shown by the extracts of *T. reesei* (C2A), *T.* sp. (C3A), and *T. spirale* (C10) strains, and only the extract of *T. spirale* (C10) indicated the occurrence of anthocyanidins. Nevertheless, resins, saponins, and amino acids were not found in any organic extracts. These results are consistent with preceding studies reporting the presence of alkaloids, flavonoids, and even phenols in *T. aureoviride* and *T. harzianum* ethyl acetate extracts [34]. 

### 2.3. GC-MS Metabolite Profile

The detected compounds in ethyl acetate extracts of the different *Trichoderma* spp. strains are presented in Table 2. The relative abundances were calculated according to the peak area percent of three replicates of each identified compound. In total, 63 metabolites were putatively identified, including terpenes, esters, fatty acids, and alcohols, among others. The extract of *T. reesei* (C2A) displayed 39 compounds, while the extracts of *T.* sp. (C3A), *T. harzianum* (C4A), and *T. spirale* (C10) showed 21, 20, and 15 compounds, respectively. All compounds appear in order of the elution of non-polar chromatographic column, and differences between extracts of *Trichoderma* strains were revealed (Figure 4). 

The sesquiterpenes were mostly found in *T. reesei* (C2A) compared to other species in which free and esterified fatty acids predominated. Sesquiterpenes are produced mainly by different plant species, but *T. virens* has been reported as the producer of these compounds with remarkable antifungal properties [35]. Nerolidol has been described for *T. reesei* and has been proven mainly as an anti-inflammatory, antinociceptive, and antioxidant compound [36]. The 1,4-cadinadiene and widdrol have been indicated to have antiparasitic [37] and antifungal [38] properties, respectively. The *β*-caryophyllene has been identified in *Trichoderma* species [39] and described as an antibacterial, antifungal, antioxidant agent [40], as well as a plant growth promoter [41]. Mevalonolactone has demonstrated impressive antibacterial activities [42]. Sorbicillin was predominantly found in *T.* sp. (C3A), followed by *T. reesei* (C2A). This polyketide has been reported in several ascomycetes, including *Trichoderma*, and possesses antimicrobial activity [43]. Indeed, a recombinant *T. reesei* strain has recently been demonstrated to improve the production of sorbicillinoids [44], and sorbicillinoids isolated from the culture filtrate of *T. ongibrachiatum* have been revealed to have antifungal effects against *Phytophthora infestans* [45]. *β*-fenchyl alcohol was detected in *T. harzianum* (C4A), and it has been reported to be an antibacterial agent [46]. Palmitic acid has been described has having antifungal activity [47] and was found in extracts of *T. reesei* (C2A), *T.* sp. (C3A), and *T. harzianum* (C4A) strains. Butyl isobutyrate has been reported in the culture filtrates of *T. asperellum* with notable cytotoxic effects against fungal plant pathogens [48] and palmitelaidic acid has also been identified as a biomarker of *T. virens* against a cotton fungal disease caused by *Rhizoctonia solani* [49], both of which were detected in extracts of *T.* sp. (C3A), *T. harzianum* (C4A), and *T. spirale* (C10). Regarding the extract of *T. spirale* (C10), ethyl linoleate has been referred to as a plant growth regulator [50], sorbic acid has been reported as an antifungal compound that impedes spore germination and mycelial growth [51], and the chiral properties of 3-hydroxybutyric acid cause it to be a mediator in the synthesis of potential antibiotics [52]. 

Notably, five metabolites were common in all strains: phenylethyl alcohol, tyrosol, diethyl succinate, and linoleic and oleic acid. Phenolic compounds are produced by another *Trichoderma* species [53], and they have been utilized as quorum sensing molecules [54] and antifungal agents [55]. In particular, tyrosol has been isolated from *T. harzianum* and *T. spirale* [56]. Furthermore, the ester has been detected in fungal endophytes [57] and shows antimicrobial activity [58], whereas fatty acids have been confirmed to have antifungal effects [47]. In fact, it has been demonstrated that unsaturated fatty acids could inhibit mycelial growth and decrease the biomass production of *M. perniciosa* [59]. 

Overall, the potential inhibition of *Trichoderma* extracts could be related to the presence of different detected extracellular metabolites due to the aforementioned biological activities. 

## 3. Materials and Methods

### 3.1. Standards and Chemicals

Potato dextrose agar (PDA) was purchased from Becton, Dickinson, and Company (Sparks, MD, USA). Gentamicin and N, O-Bis (trimethylsilyl), trifluoroacetamide (BSTFA), and dimethyl sulfoxide (DMSO) were obtained from Sigma-Aldrich (St. Louis, MO, USA). Saturated alkanes standard (C7–C40) was acquired from Supelco (Bellefonte, PA, USA). Ethyl acetate reagent grade was obtained from Fisher Scientific (Hampton, NH, USA).

### 3.2. Fungal Strains

*T. reesei* (C2A), *T.* sp. (C3A), *T. harzianum* (C4A), and *T. spirale* (C10) were provided by the Culture Collection of Microorganisms of Biotechnology Research Center of Ecuador (CCM-CIBE) [60]. These species were previously isolated from the rhizosphere of cocoa plants in Naranjal, Guayas, Ecuador. Monosporic cultures were identified by the sequencing of the internal transcribed spacer (ITS) region, as indicated by White et al. [61]. The DNA extraction and polymerase chain reaction (PCR) amplification of the universal region of fungi ITS1-5,8-ITS2 was carried out using the ITS1 (5′ TCC GTA GGT GAA CCT GCG G 3′) and ITS2 (5′TCC TCC GCT TAT TGA TAT GC 3′) primers. Subsequently, the PCR product was sequenced with the SANGER method, and the basic local alignment search tool (BLAST) was utilized for the processed sequences, from which the identities were obtained (Table 3). The fungal isolates were deposited at CCM-CIBE with (CCMCIBE-H1103), (CCMCIBE-H1104), (CCMCIBE-H1105), and (CCMCIBE-H1106) codes, respectively. In this work, the fungal spores of every strain already maintained in 10% (*v*/*v*) glycerol at −80 °C were grown individually in Petri dishes (90 mm) containing PDA culture medium with gentamicin (10 μg/mL) and incubated at 27 °C in darkness for 7 days. Then, each fungal strain was subcultured on 50 PDA plates to prepare a sufficient amount of mycelium. *Trichoderma* strains were incubated for 7 days to give the culture enough time to form robust mycelia, which are essential for the successful production of enzymes and metabolites [62].

### 3.3. Extraction Procedure

The extraction of secondary metabolites from *Trichoderma* strains followed Liu and Liu method with some modifications [62]. First, the solid medium containing fungal mycelia (SMF) of 50 inoculated PDA plates was cut into small pieces (approximately 10 × 10 mm) under aseptic conditions, transferred into a 2 L Erlenmeyer flask with 600 mL of ethyl acetate, shaken at 110 rpm for 1 h, and incubated at 25–28 °C for 24 h in darkness. Then, the organic phase was filtered, concentrated in a rotary evaporator under reduced pressure at 40 °C, and SMF extraction was repeated with recoveries every 24 h for 4 days. The extraction process was followed until the color faded, as recommended in protocols for recovering secondary metabolites [63]. Next, ethyl acetate extracts were placed together, concentrated, and dried for further analysis. This procedure was performed for each *Trichoderma* strain. The same method was used in the control treatments, except that culture media without fungi were used.

### 3.4. Antifungal Assay

Two cocoa crop pathogens preserved at CCM-CIBE were used: *M. roreri* (CCMCIBE-H815) and *M. perniciosa* (CCMCIBE-H1109). The poisoned culture medium technique was used as described by Guerrero-Rodriguez et al. [64]. The dried ethyl acetate extracts of *Trichoderma* strains were resuspended in pure dimethyl sulfoxide (DMSO) and sterilized using 0.22 μm syringe filters. After that, fungi strain disks (6 mm) were taken from 14-day-old cultures and individually transferred into Petri dishes (60 × 15 mm) containing PDA culture medium with different concentrations of *Trichoderma* extracts (1, 10, 100, and 1000 μg/mL), negative controls (DMSO and PDA), and positive controls using the commercial product Bankit^®^ 25 SC Azoxystrobin (Syngenta, Mexico City, Mexico) at 10 μg/mL. The final concentration of DMSO in Petri dishes did not exceed 1% to avoid cytotoxicity [65]. Then, four replicates of cultures were incubated at 28 °C and radial growth was measured after 2 weeks. The radial data were processed from digital images using the Image J platform. Finally, the percentage of growth inhibition (PGI) was calculated using the equation PGI = ((R1 − R2)/R1) × 100, where R1 is the radial of control and R2 is the radial of the extract [66].

### 3.5. Chemical Screening

The chemical screening tests of ethyl acetate extracts were performed as reported elsewhere [67]. Certain aliquots of *Trichoderma* ethyl acetate extracts (200 µL) were taken separately for assessment using the Dragendorff, Mayer, Wagner, Baljet, Borntrager, Liebermann–Burchard, Resins, Fehling, Foam, Ninhydrin, Shinoda, Anthocyanidins, and Catechins qualitative tests.

For the alkaloid tests, the extract was evaporated and dissolved in 1% hydrochloric acid. Then, an aliquot of acid solution of the extract was mixed with three drops of Dragendorff reagent, three drops of Wagner reagent, sodium chloride, and three drops of Mayer reagent separately. The development of turbidity confirmed the existence of alkaloids.

In the Baljet test, the extract was evaporated and dissolved in 40 µL of 100% ethanol. After that, an aliquot of 40 µL of Baljet reagent was added to the ethanol solution. The development of the red precipitate indicated the occurrence of lactones.

In the Borntrager test, the extract was evaporated and dissolved in 100 µL of chloroform. Next, an aliquot of 100 µL of 5% sodium hydroxide was added. The development of pink coloration in the aqueous phase after shaking and phases separation revealed the presence of quinones.

In the Liebermann–Burchard test, the extract was evaporated and dissolved in 40 µL of chloroform. Then, an aliquot of 40 µL of acetic anhydride and three drops of 96% sulfuric acid were added to the organic solution. The development of a red-brick reddish-brown coloration indicated the occurrence of triterpenes, and the blue coloring, which turned blue-orange, indicated the occurrence of sterols.

In the resins test, the extract was evaporated and dissolved in 100 µL of 100% ethanol. Thereafter, an aliquot of 500 µL of distilled water was added to the ethanol solution. The development of precipitates showed the existence of resins.

In the Fehling test, the extract was evaporated and dissolved in 100 µL of distilled water. Next, an aliquot of 200 µL of Fehling reagent was added to the aqueous solution and heated in a water bath for 5 min. The development of red coloration or red precipitate indicated the presence of reducing sugars.

In the foam test, the extract was evaporated and dissolved in 100 µL of 100% ethanol. Then, an aliquot of 500 µL of distilled water was added to the ethanol solution and shaken vigorously for 5 min. The presence of foam (>2 mm in height) on the surface of the liquid after 2 min revealed the occurrence of saponins.

In the ninhydrin test, the extract was evaporated, dissolved in 200 µL of 2% ninhydrin aqueous solution, and heated in a water bath for 5 min. The development of purplish blue demonstrated the existence of amino acids.

For the flavonoids tests, the extract was evaporated and dissolved in 200 µL of 100% ethanol. After that, an aliquot of 100 µL of hydrochloric acid was added to the ethanol solution. In the Shinoda test, an aliquot of 100 µL of amyl alcohol and a piece of metallic magnesium was added to the acidified ethanol solution. The development of yellow-orange-red coloration after shaking and phases separation indicated the occurrence of flavonoids. In the anthocyanidins test, an aliquot of 100 µL of distilled water and 200 µL of amyl alcohol was added to the acidified ethanol solution, and the development of red-brown coloration after shaking and phases separation revealed the presence of anthocyanidins. Finally, in the catechins test, drops of ethyl acetate extract and 20% sodium carbonate were placed on filter paper. The development of carmelite-green coloration under ultraviolet light showed the presence of catechins.

### 3.6. Gas Chromatography–Mass Spectrometry (GC-MS) Analysis

The compound separation of ethyl acetate extracts was performed in a gas chromatography–mass spectrometer from Agilent Technologies (7890A GC system and 5975C inert XL MSD with a triple axis detector). A capillary column DB-5MS (30 m × 0.25 mm) with phenyl dimethylpolysiloxane for the stationary phase (0.25 micron film thickness) and helium as the carrier gas (1.2 mL/min). The extracts were diluted in a 1:10 ratio, and injection was carried out at 250 °C with the splitless mode in triplicate. The oven temperature was initially 70 °C for 2 min, then it was increased to 300 °C at 5 °C/min and was maintained at 300 °C for 6 min. The MSD transfer line was 300 °C, and the detector temperature was 230 °C with an electron ionization of 70 eV. Data compounds were collected with the full scan mode (40–600 amu) in the quadrupole mass analyzer. Finally, metabolite identification was achieved by matching the mass spectra of the samples with the data available in Wiley 9 and NIST 2011 libraries and by comparing the linear retention indices using a series of saturated n-alkanes (C7–C40).

### 3.7. Statistical Analysis

The normalization of PGI data was performed using the arcsin transformation (arcsin √x) [68]. Thereafter, the analysis of variance (ANOVA) and Tukey’s test (*p* < 0.05) were performed to compare the inhibition of different concentrations of *Trichoderma* spp. extracts against cocoa pathogens *M. roreri* and *M. perniciosa*.

## 4. Conclusions

This study revealed that extracts obtained from the *Trichoderma* species inhibited the in vitro growth of *M. perniciosa* better than *M. roreri*. The extract of *T. spirale* (C10) showed the maximum inhibition effect at 10 µg/mL, whereas *T. reesei* (C2A) and *T. harzianum* (C4A) displayed overgrowth at higher concentrations. Differences in secondary metabolites groups and metabolite profiles that depended on the *Trichoderma* species were found. The inhibitory activity could be attributed to the antifungal compounds detected, such as widdrol, *β*-caryophyllene, tyrosol, butyl isobutyrate, sorbic acid, palmitic acid, palmitelaidic acid, linoleic acid, and oleic acid. These findings suggest that these extracts could be used to develop prospective biocontrol agents against the witches’ broom disease, especially the extract of *T. spirale* (C10). However, further chemical and antagonistic studies are required for a complete characterization of the bioactivity revealed by the extracts.

## Figures and Tables

**Figure 1 molecules-28-03208-f001:**
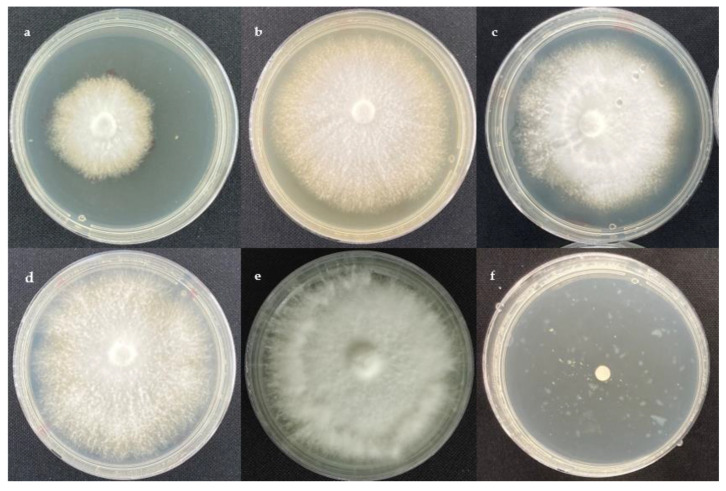
Poisoned culture assay against *M. perniciosa* after 14 days of incubation. (**a**) *T. spirale* (C10) at 10 μg/mL, (**b**) *T. harzianum* (C4A) at 1000 μg/mL, (**c**) *T.* sp. (C3A) at 100 μg/mL, (**d**) negative control DMSO, (**e**) negative control PDA, and (**f**) positive control: azoxystrobin at 10 μg/mL.

**Figure 2 molecules-28-03208-f002:**
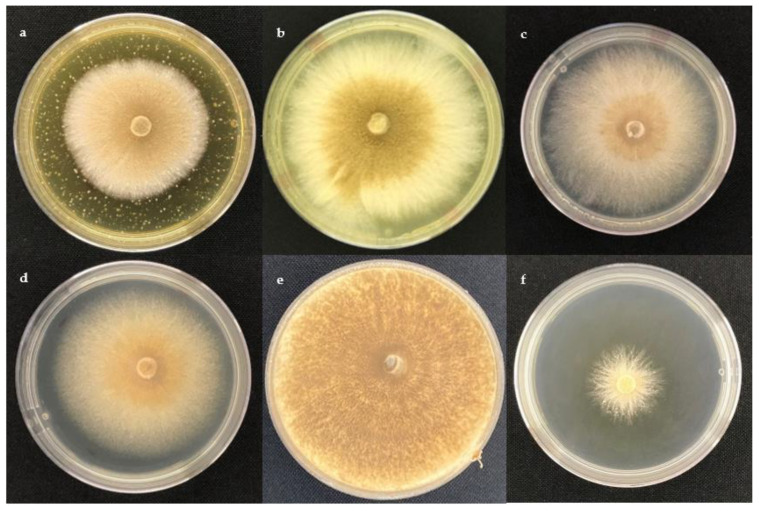
Poisoned culture assay against *M. roreri* after 14 days of incubation. (**a**) *T. spirale* (C10) at 1000 μg/mL, (**b**) *T. harzianum* (C4A) at 1000 μg/mL, (**c**) *T. spirale* (C10) at 1 μg/mL, (**d**) negative control DMSO, (**e**) negative control PDA, and (**f**) positive control: Azoxystrobin at 10 μg/mL.

**Figure 3 molecules-28-03208-f003:**
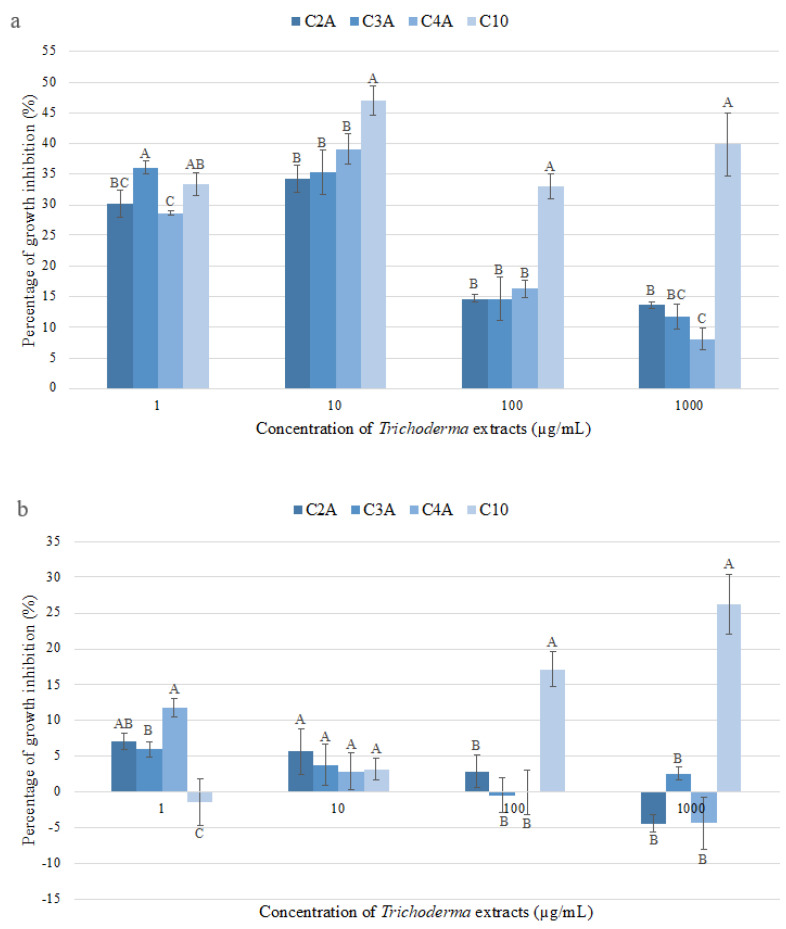
Percentage of growth inhibition (PGI) of *Trichoderma* spp. ethyl acetate extracts on cocoa pathogens (**a**) *M. perniciosa* and (**b**) *M. roreri*. Values are expressed as mean (*n* = 4) ± standard error. The same letter indicates values that are not significantly different among extracts at the assessed concentrations using Tukey’s test at (*p* < 0.05). C2A = *T. reesei*, C3A = *T.* sp., C4A = *T. harzianum*, C10 = *T. spirale*.

**Figure 4 molecules-28-03208-f004:**
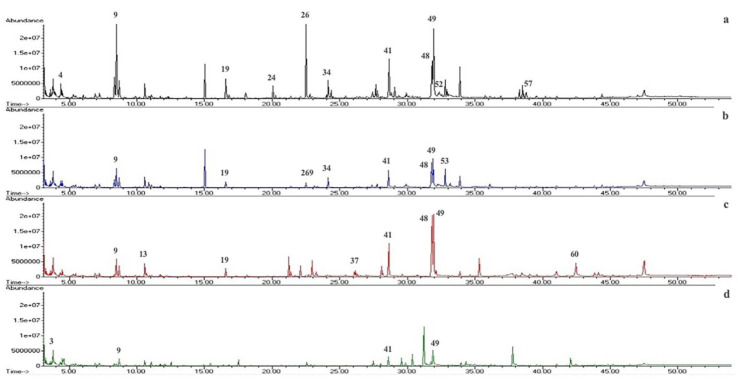
Representative chromatographic profiles of *Trichoderma* spp. ethyl acetate extracts (**a**) *T. reesei* (C2A), (**b**) *T.* sp. (C3A), (**c**) *T. harzianum* (C4A), and (**d**) *T. spirale* (C10). The major peaks have been numbered, and the corresponding compound names are given in Table 2.

**Table 1 molecules-28-03208-t001:** Chemical screening of *Trichoderma* spp. ethyl acetate extracts.

Secondary Metabolites	Test	C2A	C3A	C4A	C10
Alkaloids	Dragendorff	–	–	+	–
	Mayer	++	–	–	–
	Wagner	+++	–	–	–
Lactones	Baljet	++	++	++	++
Quinones	Borntrager	++	–	–	+++
Triterpenes and sterols	Liebermann-Burchard	+	+	+	+
Resins	Resins	–	–	–	–
Reducing sugars	Fehling	+++	+++	+++	+++
Saponins	Foam	–	–	–	–
Amino acids	Ninhydrin	–	–	–	–
Flavonoids	Shinoda	–	–	–	+
	Anthocyanidins	–	–	–	+
	Catechins	+	+	–	+

C2A = *T. reesei*, C3A = *T.* sp., C4A = *T. harzianum*, C10 = *T. spirale*. (–) absence, (+) presence, (++) abundance, (+++) high abundance.

**Table 2 molecules-28-03208-t002:** Compounds identified in *Trichoderma* spp. ethyl acetate extracts.

Peak	Compounds	Peak Area (%) ^(1),(2)^				Retention Index (Estimated) ^(3)^	Retention Index (Reference) ^(4)^
		C2A	C3A	C4A	C10		
1	Ethyl Valerate	0.54 ± 0.06	0.95 ± 0.02	-	-	901.23	884
2	3-Methylcyclohexanol	0.80 ± 0.19	-	-	-	912.21	969
3	Butyl Isobutyrate	-	1.22 ± 0.04	0.76 ± 0.04	1.37 ± 0.01	912.66	920
4	Ethyl 3-Hydroxybutyrate	1.22 ± 0.11	-	-	-	931.88	947
5	3-Hydroxybutyric acid	-	-	-	2.84 ± 0.48	942.31	938
6	1,1-Diethoxyacetone	0.05 ± 0.01	-	-	-	996.06	941
7	4-sec-Butoxy-2-butanone	-	-	0.07 ± 0.01	-	1018.54	964
8	Sorbic Acid	-	-	-	0.20 ± 0.02	1052.53	990
9	Phenylethyl Alcohol	8.90 ± 1.06	5.42 ± 0.08	3.12 ± 0.13	0.83 ± 0.01	1104.39	1136
10	5,8-Decadien-2-one,5,9-dimethyl-,(E)	-	-	0.07 ± 0.01	-	1125.63	1204
11	4-Ethoxy-4-oxobutanoic Acid	0.13 ± 0.03	-	-	-	1154.13	1141
12	Diethyl Succinate	0.11 ± 0.01	0.19 ± 0.04	0.11 ± 0.03	0.33 ± 0.00	1167.54	1151
13	β-Fenchyl Alcohol	-	-	0.69 ± 0.02	-	1186.38	1138
14	5-Hydroxy-4,4,6-trimethyl-7-oxabicyclo [4.1.0]heptan-2-one	-	-	0.36 ± 0.03	-	1200.26	1298
15	4-Hydroxy-2,4,5-trimethyl-2,5-cyclohexadien-1-one	-	-	-	0.58 ± 0.03	1235.51	1246
16	1,3-Dioxolane-2-ethanethioic acid, 2-methyl-	0.10 ± 0.03	-	-	-	1239.61	1224
17	Mevalonolactone	0.14 ± 0.02	-	-	-	1247.20	1156
18	2,5-Dimethylhydroquinone	0.04 ± 0.02	0.06 ± 0.00	-	-	1364.40	1348
19	Tyrosol	1.80 ± 0.16	1.73 ± 0.02	1.34 ± 0.07	0.19 ± 0.00	1405.59	1356
20	1,4-Cadinadiene	0.41 ± 0.02	-	-	-	1415.20	1440
21	Neoclovene	-	-	0.13 ± 0.01	-	1437.69	1416
22	Caryophyllene	0.09 ± 0.01	-	-	-	1440.57	1494
23	7-Epi-cis-Sesquisabinene hydrate	0.05 ± 0.02	-	-	-	1495.09	1523
24	Nerolidol	0.97 ± 0.07	0.11 ± 0.01	-	-	1547.13	1564
25	1-(3,3,6a-Trimethyl-1a,2,3,5,6a,6b-hexahydro-1H-6-oxa-cyclopropa[e]inden-5-yl)-ethanone	-	-	-	0.25 ± 0.02	1573.29	1508
26	Spiro[4.5]dec-8-en-7-ol, 1,8-dimethyl-4-(1-methylethyl)-	6.78 ± 0.80	1.12 ± 0.07	0.22 ± 0.05	-	1652.04	1630
27	Widdrol	0.55 ± 0.04	-	-	-	1664.67	1651
28	4-Acoren-3-one	0.30 ± 0.01	-	-	-	1672.24	1614
29	7-Phenylheptan-1-ol	-	-	-	0.21 ± 0.00	1679.42	1633
30	6-Phenylhexanoic Acid	-	0.48 ± 0.08	-	-	1679.55	1647
31	N-Methyl-N-[4-(1-pyrrolidinyl)-2-butynyl]-2-aminoacetamide	0.09 ± 0.02	-	-	-	1684.37	1770
32	Alpha-Bisabolol oxide B	-	0.32 ± 0.03	-	-	1690.13	1707
33	Formic acid, 3,7,11-trimethyl-1,6,10-dodecatrien-3-yl ester	0.09 ± 0.04	-	-	-	1690.35	1752
34	1-(hydroxymethyl)-2,5,5,8a-tetramethyldecahydro-2-naphthalenol	1.70 ± 0.18	1.99 ± 0.02	-	-	1726.09	1825
35	β-Santanol Acetate	0.06 ± 0.01	0.16 ± 0.04	-	-	1795.41	1791
36	1,1,4,6-Tetramethyl-1a,2,3,4a,5,7,7a,7b-octahydrocyclopropa[e]azulene-4,5,6-triol	0.07 ± 0.01	-	-	-	1805.00	1869
37	Methyl 5,7-hexadecadiynoate	-	-	0.32 ± 0.01	-	1828.37	1913
38	Palmitelaidic Acid	-	0.19 ± 0.03	0.53 ± 0.04	0.27 ± 0.03	1918.46	1976
39	Artemisinin	0.21 ± 0.05	-	-	-	1918.63	1903
40	(S)-3-(4-Hydroxybenzyl)piperazine-2,5-dione	0.06 ± 0.03	-	-	-	1932.11	2001
41	Palmitic Acid	4.67 ± 0.78	4.44 ± 0.04	5.93 ± 0.10	-	1942.18	1968
42	5α-Acetoxymethyl-4a,5,8,8α-tetrahydro-2,4aβ-dimethyl-1,4-naphthalindione	1.41 ± 0.13	-	-	-	1963.71	1998
43	Ethyl Palmitate	0.10 ± 0.02	-	-	-	1975.09	1978
44	2,3-Dehydro-9-hydroxy-β-agarofuran	0.26 ± 0.03	-	-	-	1979.63	2076
45	Lactaropallidin	0.36 ± 0.07	0.48 ± 0.02	-	-	2018.27	2003
46	Methyl octadeca-6,9-diynoate	-	-	0.10 ± 0.00	-	2039.21	2112
47	Hanphyllin	-	0.18 ± 0.01	-	-	2044.11	2085
48	Linoleic acid	5.00 ± 1.71	6.51 ± 0.13	13.42 ± 0.15	1.70 ± 0.14	2109.95	2183
49	Oleic acid	9.76 ± 1.80	9.63 ± 0.13	11.45 ± 0.13	7.57 ± 0.38	2116.35	2175
50	Ethyl Linoleate	-	-	-	0.48 ± 0.20	2126.39	2193
51	Cyclopropanecarboxylic acid, 2,6-di-t-butyl-4-methoxy-phenyl ester	0.61 ± 0.04	1.32 ± 0.05	-	-	2134.17	2104
52	Stearic acid	1.58 ± 0.36	-	-	-	2139.50	2167
53	Sorbicillin	1.96 ± 0.08	4.14 ± 0.06	-	-	2164.42	2160
54	3-Ethyl-3-hydroxyandrostan-17-one	-	-	0.25 ± 0.08	-	2171.72	2251
55	Acetyloxyparthenin	-	-	-	0.36 ± 0.07	2363.24	2284
56	6,8-dimethoxy-3-methyl-3-(3′-methylbut-2′-enyl)-1H-quinoline-2,4-dione	-	-	2.48 ± 0.45	-	2456.98	2397
57	S-[(E)-1,3-Diphenylbut-2-enyl] N,N-dimethylcarbamothioate	0.92 ± 0.02	-	-	-	2525.64	2436
58	4a,7a-Epoxy-5H-cyclopenta[a]cyclopropa[f]cycloundecen-4(1H)-one, 1a,6,7,10,11,11a-hexahydro-7,10,11-trihydroxy-1,1,3,6,9-pentamethyl-	0.05 ± 0.01	0.23 ± 0.02	-	-	2534.54	2591
59	1,3,5,7,9,11,13,15,17,19,21,23-Cyclotetracosadodecaene	0.16 ± 0.02	-	-	-	2619.26	2664
60	Phorbol	-	-	0.40 ± 0.06	-	2793.65	2774
61	8,9-Benzodispiro[2.0.2.4]decane, 7-(3-methoxy-2-oxa-1-oxocyclopent-5-yl)-10-phenyl-	-	-	-	0.61 ± 0.02	2952.57	2974
62	3-Hydroxyspirost-8-en-11-one	-	-	0.11 ± 0.04	-	3130.95	3044
63	7,8-Epoxylanostan-11-ol, 3-acetoxy-	0.10 ± 0.01	-	-	-	3139.42	3145

^(1)^ C2A = *T. reesei*, C3A = *T.* sp., C4A = *T. harzianum*, C10 = *T. spirale*. ^(2)^ Mean values (*n* = 3) ± standard deviation. ^(3)^ Estimated values in capillary column DB-5MS. ^(4)^ Reference values estimated in a non-polar capillary column.

**Table 3 molecules-28-03208-t003:** Summary of *Trichoderma* species identified by ITS region from the rhizosphere of cocoa plants.

Strain	Source	Location	DNA Region	Identity	Identify Code
C2A	Soil	Guayas, Ecuador	ITS1-5,8-ITS2	*Trichoderma reesei*	CCMCIBE-H1103
C3A	Soil	Guayas, Ecuador	ITS1-5,8-ITS2	*Trichoderma* sp.	CCMCIBE-H1104
C4A	Soil	Guayas, Ecuador	ITS1-5,8-ITS2	*Trichoderma harzianum*	CCMCIBE-H1105
C10	Soil	Guayas, Ecuador	ITS1-5,8-ITS2	*Trichoderma spirale*	CCMCIBE-H1106

## Data Availability

All data can be found in the manuscript.

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
