# Peer review of "Chemical Characterization of Trichoderma spp. Extracts with Antifungal Activity against Cocoa Pathogens"

_molecules, 2023, doi:10.3390/molecules28073208_

Round 1

Reviewer 1 Report

The manuscript entitled “Comparative inhibition analysis of cacao pathogens by extracts obtained from Trichoderma species” deals with the characterization of chemical compounds in the extract of four Trichoderma isolates, which exerted growth inhibition toward cocoa pathogens in vitro.

Overall, I think that the manuscript could be reconsidered for publication after major revisions, and after a deep revision of the English language (grammar and style). See the attached file for all my comments

Author Response

Reviewer 1comments

Response to reviewer comments

Report:

The manuscript entitled “Comparative inhibition analysis of cacao pathogens by extracts obtained

from Trichoderma species” deals with the characterization of chemical compounds in the extract of

four Trichoderma isolates, which exerted growth inhibition toward cocoa pathogens in vitro.

Overall, I think that the manuscript could be reconsidered for publication after major revisions.

Firstly, an accurate revision of the English style and grammar is strongly recommended. I will not

indicate the suggestions for the English language because there are too many errors. I am not an

English mother tongue, but I am expert enough to judge that there is a need for a deep revision of

the language.

The authors appreciate the reviewer's recommendations to improve the manuscript.

Then, I would suggest modifying the title, focusing more on the chemical characterization of the

extracts rather than on the growth inhibition against the pathogens, which was tested in vitro. For instance:

Chemical characterization of Trichoderma sp. extracts with antifungal activity towards cocoa

pathogens

The suggestion was considered, and changes were done in the manuscript.

Abstract

The first two lines resume too poorly the introduction. Please put the study in the right scientific

context and state the objective of the study (see also below)

The suggestion was considered, and changes were done in the manuscript.

Keywords

Eliminate endophyte, add Moniliophthora

The suggestion was considered, and changes were done in the manuscript.

Introduction

1)The first paragraph dedicated to the pathogens is poor. The authors need to enrich it with the help

of a pathologist, describing briefly how the considered pathogens infect cocoa plants, which organs

are the targets, which developmental stage etc. in order to understand how Trichoderma could

counteract the infection by the direct antifungal activity of the metabolites identified in the extract.

The suggestion was considered, and changes were done in the manuscript.

Introduction

2)The ability of Trichoderma species to induce systemic resistance in the host plants is considered

one of the main mechanisms through which this fungus exerts its beneficial activity, beyond direct

antifungal activity or mycoparasitism. Thus, please, dedicate a paragraph to “induced systemic

resistance” in the Introduction section, with adequate references.

The suggestion was considered, and changes were done in the manuscript.

Introduction

3) The objective of the study is not well-focused, because chemical characterization is the core of

this study. Thus the authors need to rephrase it. “this study addresses the chemical characterization

of the extract of four Ecuadorian Trichoderma strains which proved to inhibit the in vitro growth of cocoa pathogens etc…”

The suggestion was considered, and changes were done in the manuscript.

Figure 3 is not easily readable. I suggest making 2 figures, one for M. perniciosa and another for M.

roreri, also because the comparison between the pathogens is not correct from a statistical point of

view.

The suggestion was considered, and changes were done in the manuscript.

Figure 4 and Table 2. For more complete information, the retention time of the compounds should be given in the table, and the compound name related to the major peaks should be reported in the figure. This will facilitate the readers in recognizing the peak/compound

The Figure 1 (before Figure 4) was edited, and number of main chromatographic peaks were added. The retention index is normalized retention time in order to minimize the effects of experimental GC conditions on the magnitude of retention index.

Zellner, B. D. A., Bicchi, C., Dugo, P., Rubiolo, P., Dugo, G., & Mondello, L. (2008). Linear retention indices in gas chromatographic analysis: a review. Flavour and Fragrance Journal, 23(5), 297-314.

Line 126 Substitute “of Trichoderma spp.” with “in the ethyl acetate extracts of the different

Trichoderma spp. isolates”

The suggestion was considered, and changes were done in the manuscript.

Line 170 The sentence is interrupted. Clearly, some text is missing and cannot be evaluated at the

moment. It regards the possible use of the findings of the study, thus it is a very important piece of

text that is lacking. Please check and add the missing text when resubmitting, to permit its evaluation

The interrupted sentence was deleted. The missing text was part of the conclusions. That’s why the interrupted sentence was at the end of the “GCMS metabolite profile” section.

Line 190 Substitute “unpublished data” with an adequate reference or provide the identification

method details. In addition, and more importantly, identify the species for T. sp. C3A isolate, as

done for the other 3 isolates. Nowadays molecular biology methods allow identifying Trichoderma

at the species level.

It is important to indicate that this work was not focus on the molecular identification of Trichoderma strains. However, additional detail about the method identification were added (Line 797-802 and Table 3).

All the Trichoderma isolates were processed in the same way, but the C3A isolate was only identified up to genus. This could be explained because the ITS region used for sequencing is universal for fungi. They may not be so specific and for this reason the sequence of this isolate did not achieve a high identity some specific species. Nevertheless, further work is underway to sequence other genes and obtain more information to arrive at a better identity of it.

Line 201-203 define how long was the incubation period, the size of Petri dish, and which medium

(PDA?)

This information is in “Fungal strains” section. Line 803-809

Trichoderma strains were incubated for 7 days in order to give the culture enough time to form robust mycelia, which are essential for the successful production of enzymes and metabolites [59].

Liu, J.; Liu, G. Analysis of Secondary Metabolites from Plant Endophytic Fungi. In Plant Pathogenic Fungi and Oomycetes. Methods in Molecular Biology; Ma, W., Wolpert, T., Eds.; Humana Press: New York, NY, 2018; Vol. 1848, pp. 25–38.

Line 231-275 give the references for all the methods used

The reference used is in line 997.

Chemical screening tests of ethyl acetate extracts were performed as reported elsewhere [65]

Tiwari, P.; Kumar, B.; Kaur, M.; Kaur, G.; Kaur, H. Phytochemical Screening and Extraction: A Review. Internationale Pharmaceutica Sciencia 2011, 1, 98–106.

Line 293 Percentage data must be transformed before ANOVA. Did you? Otherwise, you need to

repeat ANOVA on transformed data and perform the Tukey test on these new data. Then you can

report percentage data in the figure, but the letters must refer to transformed data.

The suggestion was considered, and changes were done in the manuscript.

Percentage data was transformed, and Anova analysis was repeated. The new significative differences could be observed in Figure 4.

Conclusions could be written more accurately

Line 300 “bioactivity could be attributed to detected compounds” please detail which ones

The suggestion was considered, and changes were done in the manuscript.

Line 301 “strain C10 could be use as raw material” please say it better. A strain cannot be

considered a raw material

The suggestion was considered, and changes were done in the manuscript.

Line 300-302 This is true if direct antagonism is a mechanism that can counteract cocoa infection

by these pathogens. This is why I asked the authors to describe the infection in the Introduction.

Anyway, please add a sentence regarding the necessity to perform further studies to highlight also mycoparasitism ability, possible induced resistance mechanisms, as well as experiments in vivo.

The suggestion was considered, and changes were done in the manuscript.

Line 302-303 delete “and contribute relevant information about endophytic fungi from Ecuador”

The suggestion was considered, and changes were done in the manuscript.

Line 88 and 298 overgrew, overgrowth? Perhaps, did the authors mean “stimulated, stimulation”?

The word correction was considered, and changes were done in the manuscript.

Reviewer 2 Report

All the corrections are mentioned in the text. The title should be changed. Material and methods need more corrections. The discussion is incomplete and need to be revised. 

Author Response

Reviewer 2 comments

Response to reviewer comments

Report: grammar correction, suggestion and questions were placed in a pdf document of the manuscript. Questions were placed below to answer to the reviewer.

The authors appreciate the reviewer's grammar corrections and recommendations to improve the manuscript.

abundant? higher concentrations?

“The most metabolites” was changed by “the largest number of”

How do you justify?

Fungal endophytes produce several compounds such as plant growth hormones (auxins, citokinis and gibberellins), which can help also promote the growth of fungi [56].

Villavicencio, M.; Schuller, L.; Espinosa, F.; Noceda, C.; Sosa, D.; Pérez-Martínez, S. Foliar Endophytic Fungi of Theobroma Cacao Stimulate More than Inhibit Moniliophthora Spp. Growth and Behave More as an Endophytes than Pathogens. agriRxiv 2020, 2020, doi:10.31220/AGRIRXIV.2020.00019.

to what?

The interrupted sentence was deleted. The missing text was part of the conclusions. That’s why the interrupted sentence was at the end of the “GCMS metabolite profile” section.

checked?

10 mg/mL was change to 10 mg/mL

Did the incubation period 7 days? Isn't this a short time?

Trichoderma strains were incubated for 7 days in order to give the culture enough time to form robust mycelia, which are essential for the successful production of enzymes and metabolites [59].

Liu, J.; Liu, G. Analysis of Secondary Metabolites from Plant Endophytic Fungi. In Plant Pathogenic Fungi and Oomycetes. Methods in Molecular Biology; Ma, W., Wolpert, T., Eds.; Humana Press: New York, NY, 2018; Vol. 1848, pp. 25–38.

the method used by Liu and Liu is different from the method used here. It should be revised

The Liu and Liu used by method was used as reference. The sentence was edited as indicated below:

The extraction of secondary metabolites from Trichoderma strains was followed by Liu and Liu method with some modifications [59].

How did you disrupted the mycelia for extract the SM?

Mycelia disruption was not necessary since this work focused on extracellular metabolites. Thus, ultrasonic bath of ethyl acetate containing SFM was not used before reduced rotary evaporation.

Why recovery times extended for 4 days? what is the role of ethyl acetate?

During process it was noted that on the fourth day all extractables in ethyl acetate were obtained after exhaustive extraction. The extractive process was followed until the color fades as recommended protocols for recovery of secondary metabolites [60].

Kjer, J.; Debbab, A.; Aly, A.H.; Proksch, P. Methods for Isolation of Marine-Derived Endophytic Fungi and Their Bioactive Secondary Products. Nature Protocols 2010 5:3 2010, 5, 479–490, doi:10.1038/nprot.2009.233.

? obscure?

The text “indicate that strain C10 could be use as raw material for the development of a potential control agent against witches’ broom, and contribute relevant information about endophytic fungi from Ecuador” was deleted.

Reviewer 3 Report

This manuscript deals with antifungal activity of ethyl acetate extracts of Trichoderma species. The authors evaluated the activity, and further analyzed chemical components in the extracts. The examination was carried out carefully, and the results were described in detail. I think the results in this study is important on this field, however it is hard to see which parts of this paper are new findings. So, I think this manuscript should be reconstructed as below before publication.

# In Results and Discussion, the author should describe “Chemical screening”, followed by “GC-MS metabolite profile” and “Antifungal activity”

# Please discuss the results to highlight the usefulness of C10.

# In Table 2, retention index is necessary? Can Table 2 be simplified?

# The name of fungi should be expressed as only abbreviation (such as C10) through the manuscript except first part.

# Overall, “Results and discussion” should be condensed.

Author Response

Reviewer 3 comments

Response to reviewer comments

Report:

This manuscript deals with antifungal activity of ethyl acetate extracts of Trichoderma species. The authors evaluated the activity, and further analyzed chemical components in the extracts. The examination was carried out carefully, and the results were described in detail. I think the results in this study is important on this field, however it is hard to see which parts of this paper are new findings. So, I think this manuscript should be reconstructed as below before publication.

The authors appreciate the reviewer's recommendations to improve the manuscript.

# In Results and Discussion, the author should describe “Chemical screening”, followed by “GC-MS metabolite profile” and “Antifungal activity”

The suggestion was considered, and changes were done in the manuscript.

# Please discuss the results to highlight the usefulness of C10.

The suggestion was considered, and changes were done in the manuscript.

# In Table 2, retention index is necessary? Can Table 2 be simplified?

Yes, it is necessary.

The retention index is normalized retention time in order to minimize the effects of experimental GC conditions on the magnitude of retention index.

Zellner, B. D. A., Bicchi, C., Dugo, P., Rubiolo, P., Dugo, G., & Mondello, L. (2008). Linear retention indices in gas chromatographic analysis: a review. Flavour and Fragrance Journal, 23(5), 297-314.

# The name of fungi should be expressed as only abbreviation (such as C10) through the manuscript except first part.

The suggestion was considered, and changes were done in the manuscript.

# Overall, “Results and discussion” should be condensed.

The “Results and Discussion” section was improved.

Round 2

Reviewer 1 Report

The manuscript formerly entitled “Comparative inhibition analysis of cacao pathogens by extracts obtained from Trichoderma species” has been revised by the authors.

Overall, I appreciate the authors’ efforts, however, the revisions did not often completely meet my requests. I think that the manuscript is still immature, and some other work must be done before this manuscript can be published.

Notwithstanding I strongly recommended as the first point an accurate revision of the English style and grammar, I regret to say that the authors did not even try to do it. I found all the same bad language structures and grammar errors, demonstrating that no English mother tongue or professional reviewer corrected the manuscript. Also, the new text added is full of English errors. This is not acceptable since the need for English revision was already remarked.

Since the authors must meet all the reviewer’s indications, I found that this could be a reason for rejecting the manuscript.

Thus, I remark for a second time on the need for a serious revision of the English language, throughout the whole text.

Other suggestions to be considered:

Abstract

The first two lines still resume too poorly the introduction and do not say what is needed to solve the problem raised. There is a gap between the first and the second sentence that must be filled. Then I would prefer not to delete the species names in front of the code (i.e. T. reesei C2A), as in the former version.

The authors summarized in the abstract the list of metabolites found. This is not much interesting. They should highlight those metabolites that have a role as antifungal compounds, instead, and if there is a correlation with the activity found in the antifungal assay.

“Although the largest number of metabolites were produced by T. reesei C2A extract, the higher inhibition against M. perniciosa was shown by T. spirale C10”. It is not important the number of metabolites, but the quality. This sentence must be rewritten.

The last sentence of the abstract must be rephrased. It is not the finding of the work which suggests further experiments. The sentence could be “Further experiments are needed for a complete characterization of the bioactivity potential of the extracts of the different Trichoderma strains studied”  

Keywords

I asked to add Moniliophthora, while the authors did not. I ask again.

Introduction

The authors enriched the text with information about the two pathogens and the related diseases. However, I suggest completing the description of the site of infection for M. roreri, as done for M. perniciosa: this is important in order to understand how the application of Trichoderma extract could counteract the infection by the direct antifungal activity of the metabolites identified in the extract.

Results

In this revised version the authors have inverted the presentation of the chemical characterization with the antifungal assay results. I strongly recommend the authors come back to the previous version and present at first the antifungal assay results and then the chemical characterization of the extract. In fact, the last sentence of the antifungal section introduces the characterization chapter (“These results denote the importance of the identification of metabolites secreted by Trichoderma species since the extracts revealed similar properties as the aforementioned commercial antifungal agent”). Moreover, there are many sentences in the chemical characterization chapter that are related to the antifungal assay chapter, thus it is necessary to present the antifungal assay results first.

Please, always maintain the name of the Trichoderma species in the text, along with the code. It is useful and informative. Es. T. reesei C2A.

Please, put the Figure representing the chromatographic profile before Table 2, and put these figure and table nearer to the first mention in the text, not at the end of the chapter, because it is too far.

Correct the legend of the Figure like: Representative profiles of ….and add a sentence in the same legend explaining that the major peaks have been numbered and the corresponding compound names have been reported in Table 2.

Please, use differently colored histograms to identify the different extracts, inserting the related legend. This would allow leaving on the x-axis only concentration and would spare space.

Erase the legend for M. perniciosa and M. roreri in the graphs because this information is already given in the figure legend below.

Put the figure and table about antifungal assay nearer to the first mention, not at the end of the chapter, to facilitate the readers.

Statistical analysis

In this type of experiment (growth inhibition) the preferred and generally more used type of transformation of percentage data before a parametric test like ANOVA is the so-called “arcsin” transformation, i.e. arcsin (√0.50), to transform 50%. The min-max method is used in a different context.

Please, use this kind of transformation before ANOVA and Tukey test.

Conclusions

Line 391-392 I think that the authors did not understand my suggestion. Maybe I was not clear enough. I asked to add a sentence on further studies, but a conclusive sentence on the findings is necessary. The authors instead deleted the conclusive sentence. It could be “In general, these findings suggest that there is a potentiality for the use of these extracts to develop formulates for the biocontrol of the tested cocoa pathogens, especially from T. spirale C10, but further studied are needed etc…. The same for the abstract.

Again, I ask for a serious revision of the English language throughout the whole manuscript, which otherwise cannot be accepted.

Author Response

The latest suggestions have been implemented and the changes have been placed in the manuscript using track changes.

Best regards,

Reviewer 2 Report

The manuscript was corrected, but some parts need to be modified/corrected again. All the corrections are mentioned in the manuscript.

Author Response

(The authors gave the same response as above.)

Reviewer 3 Report

The manuscript was revised adequete, and I think it should be accepted in Molecules.

Author Response

The authors really appreciate the kindly comments of the reviewer.

Best regards,